# Regularized siamese neural network for unsupervised outlier detection on brain multiparametric magnetic resonance imaging: application to epilepsy lesion screening

**Zara Alaverdyan**
Univ Lyon, INSA-Lyon, Université Claude Bernard Lyon 1,
UJM-Saint Etienne, CNRS, Inserm, CREATIS UMR 5220,
U1206, F-69621, Lyon, France
zaruhi.alaverdyan@creatis.insa-lyon.fr

**Julien Jung, Romain Bouet**
Lyon Neuroscience Research Center, CRNL, INSERM U1028, CNRS UMR5292,
University Lyon 1, Lyon, France
{julien.jung, romain.bouet}@inserm.fr

**Carole Lartizien**
Univ Lyon, INSA-Lyon, Université Claude Bernard Lyon 1,
UJM-Saint Etienne, CNRS, Inserm, CREATIS UMR 5220,
U1206, F-69621, Lyon, France
carole.lartizien@creatis.insa-lyon.fr

## Abstract

Computer aided diagnosis (CAD) systems are designed to assist clinicians in various tasks, including highlighting abnormal regions in medical images. Common methods exploit supervised learning using annotated data sets and perform classification at voxel-level. However, many pathologies are characterized by subtle lesions that may be located anywhere in the organ of interest, have various shapes, sizes and textures. Acquiring a data set adequately representing the heterogeneity of such pathologies is therefore a major issue. Moreover, when a lesion is not visually detected on a scan, outlining it accurately is not feasible. Performing supervised learning on such labeled data would not be reliable. In this study, we consider the problem of detecting subtle epilepsy lesions in multiparametric (T1w, FLAIR) MRI exams considered as normal (MRI-negative). We cast this problem as an outlier detection problem and build on a previously proposed approach that consists in learning a oc-SVM model for each voxel in the brain volume using a small number of clinically-guided features [1]. Our goal in this study is to make a step forward by replacing the handcrafted features with automatically learnt representations using neural networks. We propose a novel version of siamese networks trained on patches extracted from healthy patients' scans only. This network, composed of stacked convolutional autoencoders as subnetworks, is regularized by the reconstruction error of the patches. It is designed to map patches centered at the same spatial localization to 'close' representations with respect to the chosen metric (i.e. cosine) in a latent space. Finally, the middle layer representations of the subnetworks are fed into oc-SVM models at voxel-level. The model is trained

1st Conference on Medical Imaging with Deep Learning (MIDL 2018), Amsterdam, The Netherlands.

on 75 healthy subjects and validated on 21 patients with confirmed epilepsy lesions (with 18 MR negative patients) and shows a promising performance.

# 1   Introduction

Computer aided diagnosis (CAD) systems have been introduced as to assist clinicians in various tasks such as organ or lesion segmentation, detection of abnormal regions in a medical image, etc. Recent CAD systems for brain pathologies exploit various modalities of neuroimaging data, such as magnetic resonance imaging (MRI) and positron emission tomography (PET). The vast majority of the existing CAD systems are built upon methods developed in supervised settings, using either manually designed features or currently ubiquitous deep learning architectures as in [2, 3, 4, 5]. Such systems benefit from the available data sets (usually) annotated at voxel-level, and output maps where each voxel is characterized either by a class label, a probability or, less commonly, a score discriminating healthy versus pathological voxels. Supervised learning, however, cannot be applied when the number of pathological cases in the training set is not sufficient to account for the complexity of the task. This is often the case when it comes to detecting some brain pathologies such as small vessel diseases (SVD), multiple sclerosis (MS) or epilepsy, when the lesions are subtle and vary largely in terms of shapes and textures. It is not trivial to obtain a well-annotated data set to represent such a variability. To bypass the problem of insufficient labeled data, some authors recently proposed to formulate such lesion detection tasks in semi-supervised settings, by accounting for both labeled and unlabeled data in a deep architecture for MS lesion segmentation [6] or by exploiting weak labels (the number of lesions in a scan) to detect enlarged perivascular spaces in the basal ganglia [7].

In this study we propose to tackle the problem of epilepsy lesion detection in patients with *MRI negative* exams, meaning that the lesions were not visually identified by clinicians on the MR scans. Similarly to the above mentioned lesion detection tasks, most of the current epilepsy detection methods perform supervised learning by leveraging annotated lesions delineated on *MRI positive* patients (the lesions are visually detected on the scans) [8] or by careful *a posteriori* re-reading of post-surgical scans of MRI negative patients who had undergone surgery and were seizure-free afterwards [1, 9, 10, 11]. While obtaining accurately labeled data for MRI positive patients is feasible, the real challenge is to extract accurate delineations in MRI negative patients. [11] showed that exploiting 'too generously' annotated lesions on MRI negative scans as labels for supervised learning methods leads to poor detection rate due to the presence of normal tissue in the areas labeled as pathological. Therefore, some recent methods cast epilepsy lesion detection task as an outlier detection problem [1, 11, 12]. Such an approach solely needs a training set of non-pathological images, hence no labeled data is required. [1] used a small number of features modeling the gray-white matter junction (similarly to [13, 14]) while [12] and [11] derived features from surface based morphometry (SBM). All the latter methods targeted a specific type of epilepsy caused by focal cortical dysplasia (FCD); hence the features were chosen as to provide the most common FCD-characteristics to the models.

In this work we build on the method proposed in [1] that learns a one-class SVM (oc-SVM) model for each voxel individually. Our goal is to make a step forward by replacing the handcrafted features with automatically learnt representations using neural networks. Deep learning architectures allow to learn representations that are not limited to the clinically-guided features which have to be designed for each pathology individually; the representations are learnt based on the available data. Moreover, certain architectures provide a convenient framework to learn joint representations of multiparametric/multimodality imaging. Our methodological contribution consists in proposing a variant of siamese neural network designed to learn representations for outlier detection on brain images. The network is composed of stacked convolutional autoencoders and is trained on the patches of healthy brain volumes only, by utilizing a novel loss function adapted to the given context. Such a network allows learning meaningful representations which, coupled with voxel-level oc-SVM classifiers, discriminate various brain abnormalities and can be applied to detect subtle pathologies in general. From the medical application perspective, we attempt to make a step forward in automatically learning representations for epilepsy lesion detection, unlike in the previous studies ([1, 11]). Our approach is not targeted at one specific epilepsy type and thus is more generic and also detects lesions with rather unknown signatures. Moreover, to our knowledge, this is the first study to propose a neural network architecture trained on multiparametric MRI data that can be applied to detect epilepsy lesions.

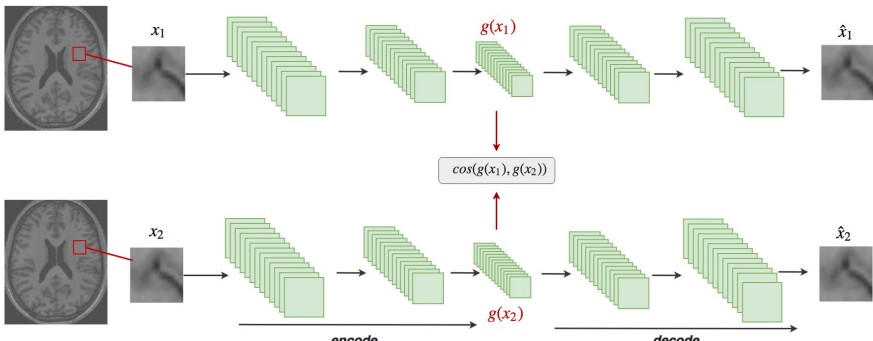

Figure 1: Siamese neural network composed of stacked convolutional autoencoders as sub-networks. The input consists of a pair of patches of 2 different subjects centered at the same spatial localization in the brain. The middle-layer representation is denoted by $g(x)$.

## 2   Method

In this study we propose to use a siamese network to learn *patch-level representations* in the context of outlier detection. Such an approach is applicable in cases where pathological samples are not available or their number is insufficient to adequately represent the nature of the pathology and hence, supervised learning is not possible.

The motivation behind the architecture choice is the following. Our objective is to map the original patches to a space where the patches belonging to different subjects but centered at the same spatial localization are "close" with respect to a chosen metric. We could consider the patches centered at the same voxel as representatives of the same class (hence, "similar" patches). In this case the number of classes would be equal to the number of voxels in a brain volume (around 4 millions) but the number of samples per class would be equal to the number of subjects. The siamese networks have proved to be efficient in similar scenarios [15, 16] where the number of classes is largely greater than the number of samples per class.

### 2.1   Regularized siamese neural network for representation learning

#### 2.1.1   Architecture

The proposed architecture is illustrated on figure 1. Our regularized siamese neural network (rSNN) consists of two identical (same architecture, shared parameters) subnetworks - stacked convolutional autoencoders (sCAE) with $K$ hidden layers and a cost module. The input $\mathbf{x}$ of a SCAE is first encoded to a middle-layer representation by a series of convolutional and max-pooling operations and later decoded with a series of deconvolutions and up-poolings to produce a reconstruction $\hat{\mathbf{x}}$ of the input. A convolutional layer $l$ is composed of $N_l$ kernels and biases and can be expressed as

$$\mathbf{H}_l^m = f(\mathbf{W}_{l-1}^m * \mathbf{H}_{l-1} + b_{l-1}^m)$$

where $\mathbf{H}_l^m$ is the $m$-th feature map of the convolutional layer $l$, $\mathbf{W}_l^m$ is the kernel matrix associated with $\mathbf{H}_l^m$ and $b_l^m$ is its bias, $f$ is an activation function (usually non-linear). $*$ denotes the convolution operation. The parameters are iteratively updated to optimize a loss function that measures the deviation between the output $\hat{\mathbf{x}}$ and the input $\mathbf{x}$.

The siamese network receives a pair of patches $(\mathbf{x_1}, \mathbf{x_2})$ at input, then each patch is propagated through the corresponding subnetwork yielding representations $g(\mathbf{x_t}), t = (1, 2)$ in the middle layer which are then passed to the loss function $L$ below. It is important to mention that, unlike in the classical siamese frameworks where the network also receives a binary label that stands for the similarity/dissimilarity of the pair, in our application all the considered pairs are *'similar'* and therefore the label is not present in the loss function. The loss function, however, can be easily modified to meet the general setting.

### 2.1.2 Loss function

Our loss function is designed to maximize the cosine similarity between $g(\mathbf{x_1})$ and $g(\mathbf{x_2})$. In the absence of dissimilar pairs (the notion of dissimilar patches is not defined in our context), it is necessary to add a regularizing term. To this end, we propose to use the mean squared error between the input patches and their reconstructions output by the subnetworks. Without a proper regularization term, the loss function could be driven to 0 by mapping all the patches to a constant value. The proposed loss function for a single pair hence is:

$$L(\mathbf{x_1}, \mathbf{x_2}; \Theta) = \sum_{t=1}^{2} ||\mathbf{x_t} - \hat{\mathbf{x}}_\mathbf{t}||_2^2 - \alpha cos(g(\mathbf{x_1}), g(\mathbf{x_2})) \tag{1}$$

where $\hat{\mathbf{x}}_\mathbf{t}$ is the reconstructed output of subnetwork $t$ of the patch $\mathbf{x_t}$ while $g(\mathbf{x_t})$ is its (vectorized) representation in the middle layer and $\alpha$ is a coefficient that controls the tradeoff between the two terms. $\Theta$ represents the parameter set.

### 2.2 Voxel-level outlier detection with oc-SVM classifiers

**A oc-SVM classifier** [17] is an outlier detection method that seeks to find the optimal hyperplane that separates the given points from the origin in a dot product space defined by some kernel function $\phi$. The corresponding optimization problem is the following:

$$\min_{\mathbf{w}, \rho, \xi_i} \qquad \frac{1}{2}||\mathbf{w}||^2 - \rho + \frac{1}{\nu n} \sum_{i=1}^{n} \xi_i \tag{2}$$

$$\text{subject to} \quad \mathbf{w} \cdot \phi(\mathbf{x_i}) \geq \rho - \xi_i, \xi_i \geq 0, i \in [1, n]$$

where $n$ is the number of training examples, $\mathbf{x_i}$ is the $i$-th example in the training dataset $X$, $\xi_i$-s are slack variables relaxing the inequality constraints as to account for the non-separable classes, $\mathbf{w}$ and $\rho$ define the separating hyperplane, $\nu$ is a parameter that sets a boundary to the fraction of outliers allowed. The decision function, then, for an example $\mathbf{x}$ is $\mathbf{w} \cdot \phi(\mathbf{x}) - \rho$. This decision function contributes to the signed score output by a oc-SVM model (in a typical scenario examples with negatives scores would be considered outliers).

To validate the usefulness of the features learnt by the proposed method, we use the *representations in the middle layer of the subnetworks ($g(\mathbf{x})$)* to train oc-SVM classifiers at voxel level. Each voxel is associated with a classifier, hence the number of classifiers is equal to the number of voxels in a volume (around 4 million voxels). For a given voxel $v_i$, the associated oc-SVM classifier $C_i$ is trained on the matrix $M_i = [\mathbf{x_{i1}}, ..., \mathbf{x_{in}}]$ where $\mathbf{x_{ij}}$ is the feature vector corresponding to the patch centered at $v_i$ of subject $j$ and $n$ is the number of subjects.

For a new patient, each voxel $v_i$ is matched against the corresponding classifier $C_i$ and is assigned the signed score output by the classifier. This yields a *distance map $D_p$* for the given patient.

### 2.3 Post-processing

For a given patient, the output of the previous step - the *distance map $D_p$*- is then post-processed to obtain the final detections. A 3-step post-processing is proposed as follows.

The first step consists in normalizing the distance maps with respect to the intra-subject spatial variability. For that purpose, the distance maps of the control subjects are computed by performing a $k$-fold evaluation of the controls in the training set (i.e. for each fold of normal subjects, the distance maps are obtained with oc-SVMs trained on the remaining subjects). These maps are used to estimate the standard deviation of the *normal subjects' distance* distribution at voxel-level. For a given patient $p$, a new map $\acute{D}_p$ is computed by a voxel-wise division of the output distance map $D_p$ over the estimated standard deviations. The final distance map $F_p$ is then derived by averaging $D_p$ and $\acute{D}_p$ i.e. $F_p = \frac{1}{2}(\frac{D_p}{max(abs(D_p))} + \frac{\acute{D}_p}{max(abs(\acute{D}_p))}))$. The reason behind the additional term is that some zones in the brain have more intra-subject variability than others and therefore are more likely to be considered as anomalies. By weighing them by the standard deviation, the score maps account for this effect.

The second step consists in thresholding the $F_p$ map to produce a *cluster map*. We keep the most negative scores up to the score corresponding to a pre-chosen *p-value* in the patient's distance score

distribution and apply a 26-connectivity rule to identify connected components which we refer to as *clusters*. The voxel clusters smaller than a fixed size (here, 82 voxels corresponding to the expected cluster size calculated with the SPM analysis of the T1 MRI data) are discarded. This allows quick elimination of small and very negative clusters which usually represent isolated intensity peaks (the size of the majority of the detected clusters varies between 500 and 1500, this threshold therefore does not affect the performance in any significant way). The *clusters* are what we refer to as *detections* by the proposed method. By varying the *p-value* the number of clusters can be controlled according to a clinician's needs.

The third step consists in ranking the detected clusters to help the analysis of the detections. For each patient individually, a *p-value* is found that produces at most 15 clusters. Among those, we use the following *ranking criterion* to assign a rank to a cluster $c_i$

$$rank(\mathbf{c_i}) \sim \lambda * \frac{score(\mathbf{c_i})}{min_j \, score(\mathbf{c_j})} + (1 - \lambda) * \frac{size(\mathbf{c_i})}{max_j \, size(\mathbf{c_j})} \qquad (3)$$

where $score(c_i)$ is the average of the voxel scores in the cluster and $size(c_i)$ is the number of voxels in the cluster. Such a ranking favors large clusters with the most negative average score. Using this ranking, we keep the top $n$ detections and discard the rest. When there is a significant overlap between a detected cluster and the ground truth for a given patient, we consider the cluster a *true positive* and *false positive* otherwise.

## 3 Experiments and results

### 3.1 Dataset description and pre-processing

The study was approved by our institutional review board with approval numbers 2012-A00516-37 and 2014-019 B and a written consent was obtained for all participants.

Our database consists of multiparametric (T1-weighted and FLAIR) MR images of 75 healthy subjects and 21 patients. They all had a 3D anatomical T1-weighted brain MRI (TR/TE 2400/3.55; 160 sagittal slices of 192 x 192 1.2mm cubic voxels) and FLAIR (176 sagittal slices of 196 x 256 1.2mm cubic voxels) on a 1.5 T Sonata scanner (Siemens Healthcare, Erlangen, Germany). All the volumes were normalized to the standard brain template of the Montreal Neurological Institute (MNI) [18] using a voxel size of 1 x 1 x 1 mm. This processing was performed using the unified segmentation algorithm [19] implemented in SPM12 also correcting for magnetic field inhomogeneities. This spatial normalisation assures a voxel-level correspondence between the subjects. We removed top 1% intensities and scaled the images between 0 and 1 at image level before feeding the patches to the rSNN.

The method has been validated on 21 patients admitted to our clinical center with confirmed medically intractable epileptogenic lesions: 2 of them were visually detected on the patient FLAIR images (but not on T1w images) and only 1 lesion was identified on both T1w and FLAIR scans. The remaining 18 patients are confirmed *MR negative* patients. The *MR negative* patients had undertaken surgeries and have been seizure-free since. The ground truth annotations used in the performance evaluation were obtained by outlining the visible zones of the *MR positive* patients and by combining the information of post-surgical MR images and the resected zones for *MR negative* patients.

### 3.2 Feature extraction with SNN

The proposed rSNN consists of two identical subnetworks - stacked convolutional autoencoders with the architecture as in fig. 2. In the mono-modal scenario, they both receive at input 15x15 patches extracted from all the available healthy subjects' volumes of the corresponding modality (T1w or FLAIR) with a stride of 8. In the multi-modal scenario, the input consists of the patches of each modality joint as channels. For each of the patches, a random 'similar pair' is found among the other subjects yielding in total around 3.5 million pairs. The $\alpha$ parameter in the loss 1 is set to 0 during the first 10 epochs, then grows linearly for 15 epochs until it reaches 0.5 and then plateaus for 5 more epochs. We used ReLU activation function in all the layers except the last one where the sigmoid is used (the input patches are scaled between 0 and 1). The Adam optimizer was used with the learning rate set to 0.001 (the rest of the parameters remained at their default value as implemented in Theano). The architecture itself is not arbitrary. The size of the patches at input was chosen after a number of tested configurations and is justified by the subtle nature of epilepsy lesions. Indeed, larger patch

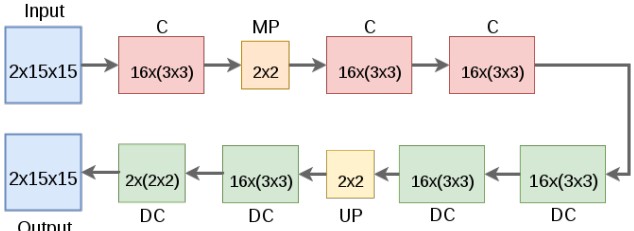

Figure 2: rSNN subnetwork architecture for epilepsy lesion detection. $C$ and $DC$ denote convolutional and deconvolutional layers respectively, $MP$ and $UP$ denote Maxpooling and Uppooling. The C and DC layers are denoted with the number of features maps (e.g. 16 for the first C layer) and the kernel size in parenthesis (e.g. 3x3 for the first C layer). With this configuration, the middle-layer is composed of 16 feature maps of size 2x2 which yields a 64-dimensional representation vector $g(\mathbf{x})$ when flattened.

sizes were not successful at detecting subtle lesions. Since the middle-layer representations are used to build oc-SVM models per voxel where the number of samples per model is equal to the number of subjects, having large representation vectors would not be beneficial. As shown on figure 2 the middle layer has 16 feature maps of 2x2 which, when flattened, yields a 64-dimensional vector.

### 3.3 oc-SVM classifier design

We used oc-SVM classifiers with RBF kernel which gives us two parameters to tune - $\nu$ (upper bound on the fraction of permitted outliers) and $\gamma$ (the kernel parameter). Varying the parameter $\nu$ did not significantly impact the results; the fraction of the outliers is controlled in the post-processing step by the threshold value applied on the distance map. It was set to 0.03 for all the voxels. The $\gamma$ parameter was derived for each voxel $v_i$ individually by estimating the median of the standardized euclidean pairwise distances of the corresponding matrix $M_i$ (see section 2.2) as in [20].

### 3.4 Results

Below we evaluate the performance of the system on 21 patients with confirmed epilepsy lesions. We first demonstrate the advantage of the multi-modal approach versus mono-modal approaches. Fig. 3 shows the true detection rates among the top $n$ clusters for 3 scenarios - voxel-level outlier detection with T1w-only, FLAIR-only and T1w/FLAIR-trained features, for 3 values of $\lambda$ of expression 3, the trade-off coefficient between the cluster size and average score. It clearly demonstrates that features learnt on the combination of multimodality data outperform the individual modalities. Moreover, the figure shows that ranking the clusters by both their average score and size has an advantage over the individual criteria. With this ranking approach, the multimodal model achieves 62% of true detections among the top 10 clusters. [11] reports a detection rate of 70% when individual SBM-based features are used; the results vary between 60 and 70% when considering combinations of some of these SBM features. 2 of the 3 *MR positive* lesions were detected among the top 2 clusters. This result is expected considering that visually detected lesions have visible markers that allow to distinguish them easily unlike the *MR negative* patients whose lesions may be detected along with other outliers of similar 'suspiciousness'.

We have also compared the global results of our CAD system to the results obtained with a general linear model (GLM) learned on feature maps derived from T1w images using three settings - 1. junction contrast, 2.extension contrast and 3. the conjunction of both contrasts - for a $p$-value of 0.001 as done in [1]. These features model the junction between gray and white matters as described in [13, 14]. For a fair comparison, the same clustering and ranking procedures (as described in section 2.3) were applied and only the top 10 clusters were considered. The results are summarized in table 1. While extension contrast detects one additional lesion compared with our mono-modal T1-based approach, the combination of junction and extension contrasts does not achieve our best performance with T1w/FLAIR model. We should also note that without applying the ranking method the original SPM implementation produces much more false positive detections without any significant change in the true positive rate.

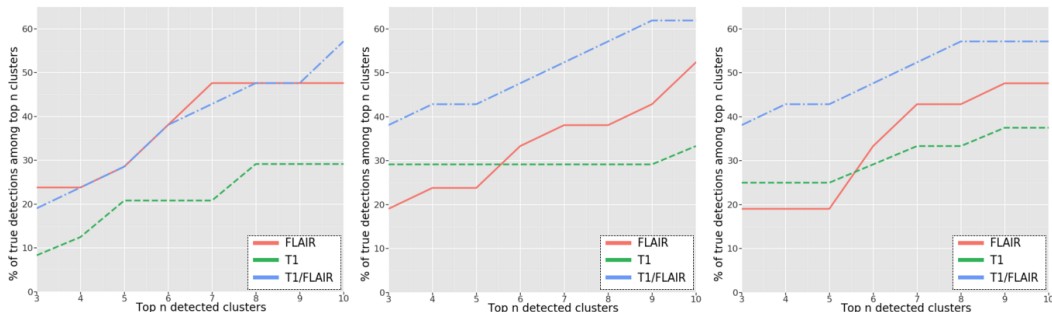

Figure 3: The performance of the CAD system. x-axis: Top $n$ clusters, y-axis: Detection rate among the top $n$ clusters. From left to right: $\lambda = 1$ (score-only), $\lambda = 0.5$ (score and size average) and $\lambda = 0$ (size-only) ranking criteria.

Table 1: Our system versus GLM model on T1w MRI as implemented in SPM software. First column: the true positive rate; the number of detected patients / total number of patients in parenthesis. Second column: the true positive rate calculated on MRI negative patients only. Third column: the average number of false positive detections per patient.

| | True positive rate | True positive rate on MR negative patients | Average # of false positives |
|---|---|---|---|
| rSNN + oc-SVM on T1 (ranked, top 10) | 0.38 (8/21) | 0.38 (7/18) | 9 |
| rSNN + oc-SVM on FLAIR (ranked, top 10) | 0.52 (11/21) | 0.5 (9/18) | 9 |
| rSNN + oc-SVM on T1/FLAIR (ranked, top 10) | **0.62 (13/21)** | **0.61 (11/18)** | **9** |
| SPM Junction on T1 (ranked, top 10) | 0.28 (6/21) | 0.27 (5/18) | 9 |
| SPM Extension on T1 (ranked, top 10) | 0.43 (9/21) | 0.44 (8/18) | 9 |
| SPM Junction-Extension on T1 (ranked, top 10) | 0.24 (5/21) | 0.22 (4/18) | 9 |

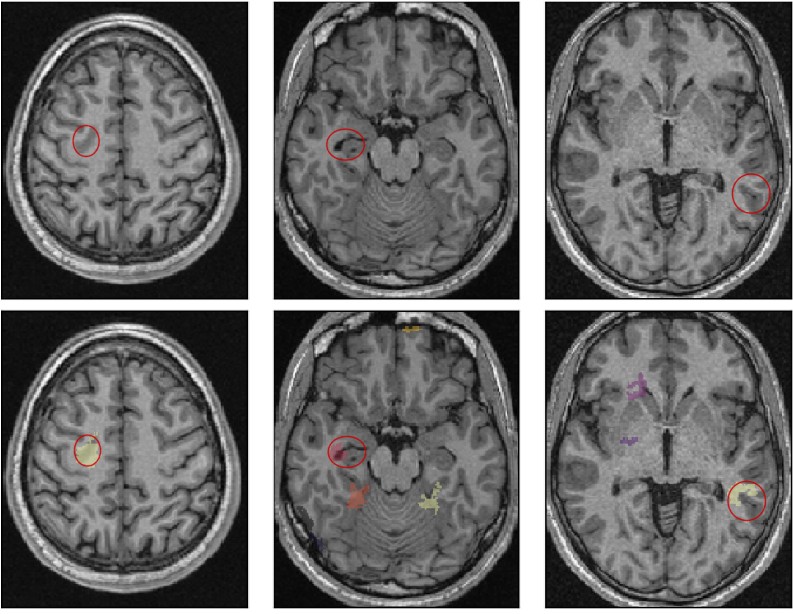

Figure 4: CAD system output for patients $A^+$, $B^-$ and $C^-$ respectively ($^+$ stands for MR positive patients, $^-$ for MR negative patients). Top row: Transverse slices centered at the lesion locations (highlighted in red circles). Bottom row: Maximum intensity projections (MIP) of the cluster maps overlaid on the MRI transverse slices. The maps show the top 1, top 6 and top 3 clusters, respectively.

# 4  Discussion

This study presents a novel method to learn representations that can be used in the task of anomaly detection on brain images. We have formulated a regularized siamese network architecture that learns normal brain representations using a set of non-pathological MR volumes. The features learnt with the network do not target specific pathology but rather allow to capture normal variability from a cohort of healthy subjects. The framework allows integrating multiple modalities and we have shown the performance gain obtained by coupling T1w and FLAIR imaging for the task of detecting subtle epilepsy lesions in MRI negative patients. To our knowledge, this is the first attempt to use deep learning for epilepsy lesion detection.

Most current studies target a specific type of the pathology, referred to as focal cortical dysplasia (FCD), mainly resulting from a malformation of cortical development and leading to drug-resistant epilepsy lesions. Some of these studies use manually designed features characterizing cortical malformations based on surface based morphometry (SBM) [9, 11, 12]. Others associate these morphometric features to the intensity anomalies in T1w MRI mainly caused by heterotopy lesions [1, 8]. Our method seeks to find more complex features in an unsupervised manner in order to identify lesions with unknown signatures. Naturally, such an approach, when applied to a specific pathology, is likely to produce more false positive detections. Although a fair comparison with published results is difficult because of the differences in the patient groups, results reported in table 1 (62% sensitivity for 9 false positives per scan) are of the same order as those reported in recent studies for the difficult task of automated detection in MRI-negative patients. Indeed, the system proposed in [11] based on SBM features coupled with semi-supervised hierarchical conditional random fields achieves 70% sensitivity on a sample of 20 T1 weighted MRI negative patients among the top 10 detections per scan. In [1], a CAD system based on morphometric and intensity features coupled with a oc-SVM classifier allows achieving the same 70% sensitivity with an average of 4 false positives per scan when evaluated on a small cohort of 8 T1w MRI negative patients.

There are different options to improve the diagnostic performance of the proposed system. First, some pathology-specific information could be introduced in the post-processing step, by discarding some of the detected clusters based on shape and/or localization criteria. In the majority of the cases, as shown in figure 4, most of the detected false positive clusters are indeed irregularities that can be easily removed by a trained radiologist. An alternative option is to move towards a semi-supervised setting by enhancing the neural network with a few 'pathological' patches that could be extracted from MRI positive cases or after a careful analysis of retrospective MRI negative patients, following, for instance, some ideas recently proposed in [21]. More improvement could be achieved by accounting for the complementary information provided by different imaging modalities. T1 and FLAIR modalities, introduced as channels to our network, allowed a significant diagnostic performance gain as shown on figure 3. We expect a further performance gain by exploiting PET imaging as recently demonstrated in [22].

Finally, the proposed method is quite straightforward to implement and to apply in daily practice as the output of the system can be obtained under a couple of minutes.

# 5  Acknowledgements

This work was performed within the framework of the LABEX PRIMES (ANR-11-LABX-0063) of Université de Lyon, within the program "Investissements d'Avenir" (ANR-11-IDEX-0007) operated by the French National Research Agency (ANR). The authors sincerely thank Valentin Hoang for his valuable contribution to the SPM analysis.

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
