# OpenReview forum: "Regularized siamese neural network for unsupervised outlier detection on brain multiparametric magnetic resonance imaging: application to epilepsy lesion screening"
_MIDL.amsterdam/2018/Conference — MIDL 2018 Oral_

### Review · AnonReviewer3 · 2018-05-07
**Solid conference paper**

**Rating:** 4
**Confidence:** 2

**Review:**

Authors present their work on identifying epileptic brain regions in MR-negative patients. The proposed unsupervised outlier detection approach is a common strategy in this field and seems promising and suitable for this task. In this work, authors have replaced the hand-crafted features with automatically learned features. The paper is clear and well-written.

Abstract: some quantitative results should be included in the abstract. Currently, the abstract is a summary of introduction/methods, but does not contain any results nor discussion/conclusion.

Introduction: authors make a valid point that currently used features are insufficient to detect all epilepsy lesions, resulting in MR-negative cases that do have epilepsy but no lesion can be found. Some description of the currently use (hand-crafted) features could be included in the introduction, to give the reader some background information on those features. These features (e.g. by Huppertz, who is not cited) are used later in the results and table 1.

Methods: authors state that their method has a high number of classes: each individual voxel in the brain. However, these are not all independent from each other, as neighbouring voxels share features/information. Is this dependence between neighbouring voxels taken into account in this method? Are the trained oc-SVM classifiers of neighbouring voxels similar to each other?

Experiments: for evaluation, it seems that authors consider an MR-negative lesions as "detected" when their method produces a detection on the pre-surgery image in the resected zone. The size of the resected zone can vary between patients and can also be quite large. How specific are the detections and how does it compare to the size of the resected zone? Is there a clear detection in the centre of the resected zone or somewhere at the border?

**Special Issue:**

Yes

---

> ### Comment · ~Zara_Alaverdyan1 · 2018-05-14
> **Thanks for the detailed feedback**
>
> We thank the reviewer for the detailed comments and remarks. Below we present some clarifications that could be included in the camera ready version.
>
> *We have referred to [1] which includes a more detailed description of the considered hand-crafted features. We agree with the reviewer in that some references could be added explicitly (including the suggested reference).
> *Yes, indeed, we could consider a simple clustering strategy as to produce 'neighborhoods' of homogeneous voxels right after the first step. In this case, the number of classifiers would be equal to the number of clusters defined in this clustering step. Instead, the number of clusters would become a parameter to tune. For now, even though homogenous voxels produce similar oc-SVM classifiers, this fact has not yet been taken into account.
> * Indeed, the resected zones may vary. The ground-truth lesions are manually outlined by experts using post-surgical scans and/or the EEG examination results. We considered a detection to be correct when it covered at least one quarter of the volume of such a 'ground truth lesion'. Detections with smaller overlaps were not considered.
> Even after the surgery, the notion of the center and boundaries of a lesion remains unclear. The patients we considered have shown definite improvement after the surgery, therefore we are sure of the resected zones; but beyond that, the characteristics of the lesions are unclear.

---

### Review · AnonReviewer1 · 2018-05-09
**Good work, just needs a few clarifications**

**Rating:** 4
**Confidence:** 2

**Review:**

This work proposed a framework for epilepsy lesion screening, posed as an outlier detection problem. This work builds on the current state of the art, which involves the handcrafting of features for specific pathologies; instead, the authors exploit the representations learned by neural networks to generalize this task. The methods are sound and the results are promising, but a few clarifications would be beneficial to the paper.

Pros:
- Sound method
- Definite improvement on the state of the art
- Novel approach to dealing with unbalanced datasets with limited labelled data
- well thought-out loss function

Cons:
- Could use more of a literature review
- Grammar revisions required

Some other comments:
- "In this case the number of classes would be equal to the number of voxels in a brain volume (around 4 millions) but the number of samples per class would be equal to the number of subjects." and ". Each voxel is associated with a classifier, hence the number of classifiers is equal to the number of voxels in a volume (around 4 million voxels)."
As the voxels in an image volume are spatially correlated with their neighbours, is there any way to exploit this property and reduce the parameters and number of classifiers required?
- Figure 4: Not sure what caption labels refer to
- How was the post-processing size threshold selected? (82) is this common in similar literature, or determined experimentally? Does this affect results at all?
- Why keep 15 clusters in post-processing?
- The "detection" metric could use some elaboration. It appears as though it's a common metric for the literature, but it is unclear for a deep learning crowd. Does it mean that it detected that a subject had a lesion? Or overlap with the ground truth lesions? Unclear.
- Figure 4: Can subjects have more than one lesion? This should be clarified. If yes, other examples with greater than one lesion should be included.

Good work, but a more thorough literature review would definitely be beneficial.

**Special Issue:**

Yes

---

> ### Comment · ~Zara_Alaverdyan1 · 2018-05-14
> **Thanks for the detailed feedback**
>
> We thank the reviewer for the detailed comments and remarks. Below we present some clarifications that could be included in the camera ready version.
>
> * Yes, indeed, we could consider  a simple clustering strategy as to produce 'neighborhoods' of  homogeneous voxels right after the first step. In this case, the number of classifiers would be equal to the number of clusters defined in this clustering step. Instead, the number of clusters would become a parameter to tune.
>
> * In figure 4 the patients are shown by columns – the left one with a visually detectable lesion, the middle and the rights ones – with invisible lesions. The second row shows the output of the system. The color bars could be improved to better visualize the clusters.
>
> * When setting the cluster size threshold, we followed the setup in [1] where this threshold corresponds to the expected size of a cluster calculated with the SPM analysis of the T1 MRI data.  The vast majority of the detected true positive clusters are much bigger (ranging between 500 and 1500 voxels) so it does not affect greatly the performance. It seems to us that this threshold could be lowered or even entirely eliminated as the final ranking criteria we consider in this study penalizes small clusters. Setting a threshold, however, eliminates faster 'very negative' and very small clusters which usually represent isolated intensity peaks.
>
> * The number of detected clusters is controlled by varying the p-value in patients' score distributions. At some point (i.e. for large p-values) clusters start to merge and become too big.  Considering greater p-values would generate an 'artificially' high detection rate. According to our observations, this happens when the number of clusters is around 15 for most of the patients. The second reason for considering only 15 clusters is that this number corresponds to a maximal detection number that could be tolerated in the clinical use.
>
> * A patient is considered 'detected' when the produced clusters overlap with the ground truth lesion by at least one quarter of their size. This will be added to the camera-ready version of the paper.
>
> *There could be patients with multiple lesions but our data set consists of one-lesion patients only.

---

### Review · AnonReviewer2 · 2018-05-09
**Regularized siamese neural network for unsupervised outlier detection on brain multiparametric magnetic resonance imaging: application to epilepsy lesion screening**

**Rating:** 3
**Confidence:** 2

**Review:**

Summary
This paper tackles the problem of detection of subtle lesions in MRI formulated as an outlier detection approach applied to the problem of epilepsy lesion detection.
The proposed approach tries to overcome the need of manual annotations for supervised approaches by only using non-pathological cases for training, which allows unsupervised learning.
The proposed method is an improvement of previous approach, based on SVM classifier, but now hand-crafted features are replaced with a representation learned with neural networks.
Siamese network is modified by removing the label "similar/dissimilar", because all patches are "similar" in the designed approach.
Therefore, the loss function is modified to include the cosine similarity of encoded representation and the mean squared error between input and output patches.

Strong points
* Exploiting data in an unsupervised fashion.
* Authors claim this is the first time this approach is applied to multiparametric MRI data.
* The modified loss function is interesting, and allows to avoid learning wrong representation that fullfill the loss in a trivial way (all zeros, as mentioned in the paper).

Weak points
* Underlying idea is not novel as is based on previous work.
* What is the main difference between the proposed loss function and other loss functions used in autoencoders? Isn't the different between input and reconstructed patch considered already?
* A threshold of 82 voxels is used, but not justified. Why?
* Same for the 15 clusters considered. Why 15?
* How sensitive is this method to the registration part? How can a voxel-wise correspondence be ensured? It seems that this method stronglty relies on the output of SPM. Could other pipeline be used, and how is the method robust to the use of other systems (in case for example SPM is not available or not allowed, for example in a clinical setting)?
* Image rescaling between 0 and 1 is done at image level, or statistics on the training set are used?
* In Figure 4, case B and C have clusters in positions that are close to the lesion, but not containing lesion voxels, apparently.
* The performance of SVM with hand-crafted feature is reported to be 70% in [11]. Does this mean that the proposed approach under-perform the previous one? Are the datasets used in both papers the same?

Extra comments
* Having an embedded represteation of 2x2, still keeps spatial information. Other papers have a 1x1xN feature map in this position of the encoder. What would be the difference?
* Could a semi-supervised approach be used here as well? It seems to work well when hand-crafted fetaures are used, so it could boost the eprformance of this approach, if some supervision is introduced.

**Special Issue:**

No

---

> ### Comment · ~Zara_Alaverdyan1 · 2018-05-14
> **Thanks for the detailed feedback**
>
> We thank the reviewer for the detailed comments and remarks. Below we present some clarifications that could be included in the camera ready version.
>
> * A typical loss in an autoencoder-like setting mainly accounts for the 'reconstruction' task as to reproduce the input from its compressed representation. When there are no additional constraints  in the loss function, it does not impose  explicit 'similarity' in the latent space; the 'similarity' is rather implicit i.e. when input patches are similar enough, so will be their representations. In this method we seeked to impose the latent-space similarity as the reconstruction error alone was not enough.
> Translating to the considered medical context, since there is a normal between-subject anatomical variability, our aim was to design a method that, when given two non-pathological but naturally varying patches, maps them 'closely' in the representation space. In this way, outlier/abnormal patches are detected better in the following oc-SVM step.
> * When setting the cluster size threshold, we followed the setup as in [1] where this threshold corresponds to the expected size of a cluster calculated with the SPM analysis of the T1 MRI data.  The vast majority of the detected true positive clusters are much bigger (ranging between 500 and 1500 voxels) so it does not affect greatly the performance. It seems to us that this threshold could be lowered or even entirely eliminated as the final ranking criteria we consider in this study penalizes small clusters. Setting a threshold, however, eliminates faster  'very negative' and very small clusters which usually represent isolated intensity peaks.
> *The number of detected clusters is controlled by varying the p-value in patients' score distributions. At some point (i. e. for large p-values) clusters start to merge and become too big.  Considering greater p-values would generate an 'artificially' high detection rate. According to our observations, this happens when the number of clusters is around 15 for most of the patients. The second reason for considering only 15 clusters is that this number corresponds to a maximal detection number that could be tolerated in the clinical use.
> *Two registration methods implemented in SPM (DARTEL and unified segmentation) were tested with the performance being similar. We did not compare more methods which could be done in the future. Any registration technique, implemented in any other toolbox, should be compatible with the approach. We are aware that the registration may not be perfect; we also think that this effect is alleviated in the proposed normalisation of the output score maps, consisting in re-weighting the output maps by the standard deviations at voxel level.
> * The rescaling was done at image level.
> *In cases B and C, the lesions were not visually detected by clinicians. The lesions are roughly outlined by the red circles on figure 4.  As indicated in the paper, the ground truth for these patients was manually annotated by expert clinicians based on post-surgical MR images and the resected zones. The detected clusters achieve a significant overlap with the ground truth.
> * The data set in [11] is not the same as ours. Since [11] exploits hand-crafted features, clinically-derived to characterize a certain type of epilepsy (FCD), it achieves better results on their data set consisting of the corresponding epilpesy-type-patients. It seems to us that our data set is more heterogeneous since it is not only limited to FCD lesions.
>
> *We have not tried the 1x1xN setting in the middle layer  but  it definitely could be interesting to test in the future.
> * The problem with a semi-supervised approach remains the accurate annotation of the lesions in MR negative patients. As a perspective to this work, we consider adding some supervision from MR positive patients where the lesions are visible. This could favor 'easily detectable' lesions over the 'invisible' ones. Hand-crafted features could be plugged in as well to tailor the CAD system to a specific type of lesions, for instance.

---

### Decision · Program_Chairs · 2018-05-15
**Paper50 Acceptance Decision**

Oral